# Bacterial Aetiology of Neonatal Sepsis and Antimicrobial Resistance Pattern at the Regional Referral Hospital, Dar es Salam, Tanzania; A Call to Strengthening Antibiotic Stewardship Program

**DOI:** 10.3390/antibiotics12040767

**Published:** 2023-04-16

**Authors:** Mtebe Majigo, Jackline Makupa, Zivonishe Mwazyunga, Anna Luoga, Julius Kisinga, Bertha Mwamkoa, Sukyung Kim, Agricola Joachim

**Affiliations:** 1Department of Microbiology and Immunology, School of Medicine, Muhimbili University of Health and Allied Sciences, Dar es Salaam P.O. Box 65001, Tanzania; agricolaj@yahoo.com; 2Medipeace Global Health, Dar es Salaam P.O. Box 77978, Tanzania; jackline.makupa@medipeace.org (J.M.); sukyung.kim@medipeace.org (S.K.); 3Mwananyamala Regional Referral Hospital, Dar es Salaam P.O. Box 61665, Tanzania

**Keywords:** neonatal sepsis, bacterial aetiology, antimicrobial resistance, multidrug-resistant bacteria, antibiotic stewardship

## Abstract

The diagnosis of neonatal sepsis in lower-income countries is mainly based on clinical presentation. The practice necessitates empirical treatment with limited aetiology and antibiotic susceptibility profile knowledge, prompting the emergence and spread of antimicrobial resistance. We conducted a cross-sectional study to determine the aetiology of neonatal sepsis and antimicrobial resistance patterns. We recruited 658 neonates admitted to the neonatal ward with signs and symptoms of sepsis and performed 639 automated blood cultures and antimicrobial susceptibility testing. Around 72% of the samples were culture positive; Gram-positive bacteria were predominantly isolated, contributing to 81%. Coagulase-negative *Staphylococci* were the most isolates, followed by *Streptococcus agalactiae*. Overall, antibiotic resistance among Gram-positive pathogens ranged from 23% (Chloramphenicol) to 93% (Penicillin) and from 24.7% (amikacin) to 91% (ampicillin) for Gram-negative bacteria. Moreover, about 69% of Gram-positive and 75% of Gram-negative bacteria were multidrug-resistant (MDR). We observed about 70% overall proportion of MDR strains, non-significantly more in Gram-negative than Gram-positive pathogens (*p* = 0.334). In conclusion, the pathogen causing neonatal sepsis in our setting exhibited a high resistance rate to commonly used antibiotics. The high rate of MDR pathogens calls for strengthening antibiotic stewardship programs.

## 1. Introduction

Neonatal sepsis is a clinical syndrome in infants within the first 28 days of life, manifested by systemic signs and symptoms of infection and isolating a bacterial pathogen from the bloodstream [1,2]. Early-onset neonatal sepsis develops within the first 72 h of life after birth, and late-onset develops after 72 h [3]. Early onset infection is usually due to vertical transmission by ascending contaminated amniotic fluid or during vaginal delivery from bacteria in the mother’s genital tract [4,5]. Late-onset sepsis is caused by bacteria acquired from the healthcare environment, community, and vertical transmission after initial neonatal colonisation that evolves into a later infection [3,6].

Signs and symptoms of neonatal sepsis are multiple and require the presence of two or more to make a clinical judgment. Diagnosing and managing neonatal sepsis, especially early onset, are significant challenges due to its nonspecific signs and symptoms [7]. The clinical presentation of neonatal sepsis varies, and no pathognomonic features exist [8]. In many resource-limited settings, clinicians make tentative diagnoses and administer treatment based on signs and symptoms [9]. However, the aetiology of neonatal sepsis and the response to empirical antimicrobial agents vary significantly, seasonally, geographically, or at the level of healthcare facilities [8,9,10]. The Society of Critical Care Medicine, the European Society of Intensive Care Medicine, and the International Sepsis Forum recommend empiric broad-spectrum therapy with one or more antimicrobials to cover all likely pathogens, narrowing once the pathogen(s) and sensitivities are available [11]. Ampicillin, cloxacillin, and gentamycin have been widely used as first-line antibiotics for empirical treatment [9,10]. Some studies have reported that treatment with ampicillin combined with gentamicin or cefotaxime is effective [12]. However, an alarming rate of emerging antibiotic-resistant strains of common antibiotics poses a challenge to empirical treatment [10,13,14,15]. The guideline for managing neonatal sepsis in our setting requires initiating empiric treatments with a first-line regimen comprising ampi-clox (ampicillin and cloxacillin fixed-dose combination) and gentamicin. If there is no clinical improvement in vital signs within 72 h, the treatment protocol requires a switch to ceftriaxone as a second-line regimen.

Following the dynamic nature of the aetiological agents and susceptibility to antimicrobial agents, there is a need to periodically update the local data to determine the choice of antibiotics for neonatal sepsis. Therefore, we designed the hospital-based cross-sectional study to determine the aetiology and antibiotic susceptibility profiles of bacteria causing neonatal sepsis at a Regional Referral Hospital in Dar es Salaam, Tanzania. Data from this study may be used in drafting evidence-based empirical treatment guidelines for neonates where blood cultures for neonatal sepsis are not feasible.

## 2. Results

### 2.1. Patients’ Demographic and Clinical Characteristics

The study recruited 65h8 neonates with features of neonatal sepsis, aged between 0 and 28 days, with a median age of 2 [Interquartile range (IQR): 1–8] days. Most neonates, 401 (60.9%), were aged up to three days; 348 (52.9%) were female; and 561 (85.3%) had an average birth weight of 2500 g or above. An almost equal number of neonates were born at Mwananyamala Regional Referral Hospital (MRRH) or other facilities within the district. Around 17.5% of neonates were delivered through the cesarian section, and 16.0% were born prematurely at a gestation age of fewer than 37 weeks. Of the neonates admitted, 405 (61.6%) had a clinical diagnosis of early neonatal sepsis, 50 (7.6%) had birth asphyxia, and 104 (16.4%) used antibiotics before admission. Fever was the most common complaint reported in 315 (83.3%), while 247 (37.5%) had more than two clinical features. Empirical therapy was provided for all neonates before laboratory results; 592 (82.4%) received only the first-line regimen, while 116 (17.6%) switched to the second-line regimen. The mortality rate was 4.3% (28/658); in addition, 5.8% (38/658) neonates were referred to a tertiary hospital for further management (Table 1).

### 2.2. Laboratory-Confirmed Neonatal Sepsis and Distribution of Causative Pathogens

Of the 658 neonates recruited, blood cultures were performed in 639 (97.1%), of which 460 had bacterial growth, which gave an overall laboratory-confirmed neonatal sepsis of 72% [95% CI: 68.3–75.4]. Gram-positive bacteria were the predominant pathogens, counting 374 (81.3%). Most isolates, 163 (35.4%), were Coagulase-negative *Staphylococcus* species (CoNS), followed by *Streptococcus agalactiae* (*S. agaractiae*), 99 (21.5%), and *Staphylococcus aureus* (*S. aureus*), 94 (20.4%). *Escherichia coli* (*E. coli*) was the most isolated Gram-negative bacteria, accounting for 42/460 (9.1%) of the total isolates and 48% (42/86) of Gram-negative bacteria (Table 2).

### 2.3. Antibiotic Resistance by Pathogens Causing Neonatal Sepsis

Gram-positive bacteria demonstrated an overall proportion of antibiotic resistance ranging from 23.1% (chloramphenicol) to 92.9% (penicillin). Among Gram-positive bacteria, a high antibiotic resistance rate (above 50%) was demonstrated for most of the antibiotics tested except chloramphenicol (23.1%), clindamycin (39.9%), and ciprofloxacin (47%) (Table 3). For Gram-negative bacteria, the overall percentages of antibiotic resistance ranged from 24.7% (amikacin) to 90.7% (ampicillin). In addition to ampicillin, a high antibiotic resistance rate (above 50%) was shown towards trimethoprim-sulfamethoxazole (86.6%), amoxicillin-clavulanic acid (88.2%), piperacillin (63.5%), and ceftazidime (76.5%). *E. coli*, the predominant Gram-negative bacteria isolated in this study, had an antibiotic resistance rate ranging from 16.7% (piperacillin-tazobactam) to 92% (trimethoprim-sulfamethoxazole) (Table 4).

### 2.4. Empirical Treatment and Antibiotic Resistance Pattern

The results of antibiotic susceptibility tests show that only 198 (43.6%) neonates received appropriate 1st line empirical therapy (ampicillin + cloxacillin+ Gentamycin). Among the neonates with antibiotic susceptibility results, only 9 (25.7%) out of 35 were appropriately switched to 2nd line empirical therapy, and 44.4% (165/377) appropriately remained on 1st line empirical therapy (Table 5).

### 2.5. Prevalence of Multidrug-Resistant Strains Causing Neonatal Sepsis

The overall prevalence of multidrug-resistant (MDR) strains of bacteria causing neonatal sepsis was 70.4%, non-significantly more among Gram-negative bacteria (64, 75.3%) than in Gram-positive bacteria (243, 69.2%) (*p* = 0.008). All isolates except *S. agalactiae* showed resistance to five antibiotic classes. The proportion of MDR strains ranged from 40.8% in *S. agalactiae* to 100% in *Enterobacter cloacae*. and *Serratia marcescens*. All isolated pathogens except *S. agalactiae* had a proportion of MDR strains ranging from 76.2% to 100% (Table 6). Regarding the clinical outcome, death occurred in 10 (3.2%) compared to 7 (5.0%) neonates infected with non-MDR strains; however, the difference was not significant (*p* = 0.371).

## 3. Discussion

The current study determined the aetiology of neonatal sepsis and the antibiotic resistance profiles among neonates admitted to a regional referral hospital in Dar es Salaam, Tanzania. The study revealed a predominance of Gram-positive bacteria among pathogens causing neonatal sepsis. In addition, the study observed a relatively high rate of multidrug-resistant gram-negative and positive bacteria causing neonatal sepsis, which implies significant antibiotic resistance challenges.

Nearly eight out of ten isolated pathogens causing neonatal sepsis were Gram-positive bacteria, of which CoNS were frequently isolated, followed by *S. agalactiae***.** Some studies in Tanzania also reported that Gram-positive bacteria were often isolated in neonatal sepsis compared to Gram-negative bacteria, with *S. aureus* being the most isolated pathogen [10,16]. In contrast, other studies reported Gram-negative organisms as the common cause of neonatal sepsis, including *Klebsiella* spp., *E. coli*, *Pseudomonas* spp., and *Salmonella* spp. [17,18] Collectively, our study found mixed pathogens: those transmitted from the mothers during delivery [19] and those acquired from the healthcare environment or community [6]. The findings may reflect the low quality of obstetric and neonatal care.

Pathogens that cause neonatal sepsis vary according to time of onset, geographical differences, and source of infection [5,6,20]. A report from the United States indicates prospective surveillance at Eunice Kennedy Shriver National Institute of Child Health and Human Development Neonatal Network centres indicates Group B *Streptococci* (GBS) and *E. coli* were the most common pathogens in early-onset neonatal sepsis [21]. A study conducted in Ethiopia showed the most common pathogens causing neonatal sepsis to be GBS, *S. aureus*, and *E. coli* [22], while another study in southeast Ethiopia found CoNS, *E. coli*, and *S. aureus* to be the common causes of neonatal sepsis [23]. In Nigeria, *Klebsiella* spp. was reported as the predominant microorganism causing neonatal sepsis [24].

In the current study, CoNS accounted for 35.4% of laboratory-confirmed neonatal sepsis. Some consider CoNS contaminants and sometimes do not include them in the analysis due to a lack of proof that they are true pathogens [25]. One study found CoNS to be a true pathogenic infection, contributing to more than half of late-onset neonatal [26]. Even though we did not prove the clinical significance of isolated CoNS, we included them in the overall analysis of neonatal infection due to increasing evidence that they were true pathogens [27,28]. However, it is also likely that the apparent isolation of CoNS in early-onset neonatal sepsis may reflect contamination, as CoNS are associated more with late-onset neonatal sepsis [23,26].

In the current study, *S. agalactiae* was the second most frequently isolated pathogen causing neonatal sepsis, contributing to 21% of the isolates. Other studies also reported *S. agalactiae* as the most common aetiology of neonatal sepsis [29,30,31]. In contrast, *S. agalactiae* has rarely been reported in other studies conducted in Tanzania [10,16,32,33]. The difference may reflect variations in risk factors and the change in the aetiology of neonatal sepsis with time. *S. agalactiae* asymptomatically colonises the lower genital and gastrointestinal tracts and colonises 10–35% of pregnant women [34], which may be transmitted to neonates during delivery and cause neonatal infection [5]. Indeed, a previous study demonstrated a clear link between maternal GBS colonisation and an increased risk of early neonatal sepsis [35]. The status of *S. agalactiae* neonatal infection found in our study calls for initiating clinical intervention to prevent neonatal sepsis by intrapartum antibiotic prophylaxis [36].

Our study found high antibiotic resistance in first- and second-line empirical treatment of neonatal sepsis. *E. coli* and *Klebsiella* were highly resistant to ampicillin, gentamycin, and ceftriaxone; *S. aureus* and CoNS were highly resistant to oxacillin; and *S. agalactiae* were highly resistant to ceftriaxone. Over 80% of *E. coli* and *Klebsiella* isolates were resistant to ampicillin, while around 60% of *S. aureus* and CoNS were resistant to oxacillin. Resistance to gentamycin was 66% and 47% for Gram-positive and Gram-negative pathogens, respectively. Notably, the antibiotic resistance of Gram-positive pathogens was relatively low (less than 40%) to chloramphenicol and clindamycin, while for Gram-negative pathogens, a low resistance rate was found to amikacin, piperacillin/tazobactam and chloramphenicol. A point prevalence study in European countries reported that 40% of pathogens isolated were resistant to first-line antibiotics prescribed by the WHO [37]. The high resistance shown by the bacterial pathogen in the current study may be attributed to the excessive and irrational use of these antibiotics in our setting [38,39]. Our findings show that the guideline for empirical antibiotic therapy in neonatal sepsis led to around 56% inappropriate therapy considering the result of susceptibility testing of causative organisms. Our findings of high levels of resistance to first-line antibiotics are similar to those of other studies [10,16,40,41], highlighting the need to review the current treatment guidelines and implement antibiotic stewardship. We advocate for robust surveillance of antibiotic resistance linked to clinical and treatment data to inform the rational use of antibiotics.

The proportion of MDR strains among patients with neonatal sepsis was remarkably high for both Gram-positive and Gram-negative pathogens. However, the proportion of MDR strains was higher in Gram-negative than Gram-positive bacteria. The current findings are comparable to results from other studies [42,43,44,45]. An Ethiopian study reported an equal proportion of MDR strains of similar pathogens causing neonatal sepsis [42]. A systematic review of five countries in South Asia showed that Gram-negative bacteria were MDR in 54% to 79% of isolates [43]. One study in Jordan found that 69% of Gram-negative organisms were MDR strains [44]. A study in China reported that around 50% of Gram-negative organisms were resistant to third-generation cephalosporins [45]. A high proportion of MDR strains among pathogens causing neonatal sepsis calls for implementing antibiotic stewardship and AMR surveillance in neonatal infection. As reported, one of the main factors for the emergence of AMR is the irrational use of antibiotics in empiric treatment [46], which is the practice in our setting.

We acknowledge a limitation on the laboratory methods for the differentiation of CoNS. In addition, Anaerobes that may cause sepsis were not investigated. Finally, CoNS isolated in early neonatal sepsis may be due to contamination during blood collection, as we used one set of blood cultures.

## 4. Materials and Methods

### 4.1. Study Design and Setting

This cross-sectional study was conducted from January 2021 to July 2022 at MRRH in the Dar es Salaam region of Tanzania. The hospital serves a population of around 2,226,692 in the Kinondoni District. It has a bed capacity of 254 and a bed occupancy of 87%. The hospital offers reproductive and child health services, including antenatal, postnatal, and newborn care. The average neonatal admission in a month is 200.

### 4.2. Study Population

We enrolled neonates aged 0–28 days admitted to the neonatal ward. We included only neonates who had at least two clinical features suggestive of sepsis [47]. The features included an axillary temperature of <36.5 °C or >37.5 °C, jaundice, a respiratory rate of <30 or >60 breaths per minute, severe respiratory distress (grunting, central cyanosis, hypoxia (SpO_2_ < 90%), poor feeding, a random blood sugar of <2.5 or >11.0 mmol/L, umbilical cord or skin infection, a pulse rate of <100 or >160 beats per minute, irritability, seizures, lethargy, and altered consciousness. Neonates whose mothers were critically ill and failed to consent for their neonates to be enrolled were excluded.

### 4.3. Data and Sample Collection

We used a structured data collection tool (designated neonatal infection assessment and reporting tool) to collect data for demographic and clinical characteristics by interviewing the mother of each participating neonate. In addition, physical measurements were performed for each neonate during the neonatal clinical assessment. Blood samples for culture were collected by well-trained healthcare workers from the peripheral veins into commercial blood culture bottles (bioMerieux SA, Lyon, France) for paediatrics in a ratio of 1:5 before administering antibiotics. The aseptic technique was adhered to during sample collection to avoid contamination. A laboratory requestion form was used to record specimen information and laboratory findings.

### 4.4. Laboratory Procedures

#### 4.4.1. Isolation and Identification

The inoculated blood culture bottles were incubated at 36–37 °C in the automated blood culture machine, the BACT Alert Microbial Detection System (bioMerieux SA, Lyon, France). Blood culture was considered positive if flagged within five days of incubation. Positive blood cultures were sub-cultured on plates of 5% sheep blood supplemented Columbia Blood agar (BA) (Oxoid Ltd., Chester, UK) and MacConkey agar with crystal violet (MCA) (Oxoid Ltd., Chester, UK). Inoculated plates were incubated at 35 ± 2 °C for 24 h. Microbial colonial characteristics on a culture medium were documented. The pure isolates obtained from subculture plates were identified through microbiological techniques, including colony morphology, Gram-staining reactions, and biochemical tests. The biochemical tests included catalase, coagulase, DNase, oxidase, Sulphur, Indole, Motility (SIM), Kligler Iron Agar, citrate utilisation tests, and lactose fermentation. Analytical Profile Index (API) 20E and API 20-NE (bioMérieux SA, Lyon, France) tests were also used in cases of inconclusive conventional biochemical test results.

#### 4.4.2. Antibiotic Susceptibility Testing

Antimicrobials Susceptibility Testing (AST) was performed using the Kirby-Bauer disk diffusion method [48]. Inhibition zones were interpreted as recommended by the Clinical and Laboratory Standards Institute (CLSI) guidelines, 2020 [49]. Briefly, bacterial colonies from pure culture were transferred to a tube containing 5 mL of sterile 0.9% normal saline, then gently mixed to form a homogenous suspension equivalent to 0.5 McFarland standard solution. A sterile cotton swab was dipped into the bacterial suspension, and the excess fluid was removed by gently pressing and rotating the swab against the inside wall surface of the tube. The swab was then used to inoculate the bacteria evenly over the entire surface of the Mueller-Hinton agar (MHA) (Oxoid Ltd., Chester, UK) plate. MHA plates were incubated at 35 ± 2 °C incubator for 16–18 h. For *Streptococci* spp. MHA plates supplemented with 5% sheep blood were set in a candle jar at 35 ± 2 °C for 16–18 h. Diameters of the zones of inhibition around each antibiotic disk were measured with a ruler in millimetres.

Ampicillin (10 µg), trimethoprim-sulfamethoxazole (25 µg), ciprofloxacin (5 µg), gentamicin (10 µg), clindamycin (2 µg), erythromycin (15 µg), oxacillin (5 μg), penicillin (2U) chloramphenicol (30 μg), ceftriaxone (30 µg) (for *Streptococci*), and linezolid (10 µg) (Liofilchem, Italy) were tested for Gram-positive pathogens. Whereas ampicillin (10 µg), trimethoprim-sulfamethoxazole (25 µg), amoxicillin-clavulanic acid (30 µg), ciprofloxacin (5 µg), ceftazidime (30 µg), ceftriaxone (30 µg), gentamicin (10 µg), chloramphenicol (30 µg), meropenem (10 µg) amikacin (30 µg), Piperacillin (30 µg), and Piperacillin/Tazobactam (40 µg) (Liofilchem s.r.l, Roseto Degli Abruzzi, Italy) were used for Gram-negative pathogens. The isolates showing resistance to one agent in at least three different classes of antibiotics were defined as MDR [50].

#### 4.4.3. Quality Control Measures

The culture media were prepared according to the manufacturer’s instructions. A sterility check was performed every time a new batch of media was prepared by incubating a sample of 5% of the prepared media at 37 °C for 48 h. The ability of media to support the growth of the suspected organisms was determined by inoculating the medium with a typical stock culture. Negative and positive controls were used to determine the biochemical response of the reagents/test kits. We used reference strains *Escherichia coli* ATCC 25922, *Klebsiella pneumonia* ATCC 700603, and *Staphylococcus aureus* ATCC 25923 for the quality control of the culture media, discs, biochemicals, AST, and incubation conditions.

#### 4.4.4. Data Analysis

We entered the data into the EPI info software and analysed it at various levels using the statistical package for the social sciences (SPSS) version 26 (Armonk, NY, USA: IBM Corp). Categorical data are presented in percentages and fractions, while continuous data are presented in the median [IQR; Interquartile ranges]. The Chi-square test was used to determine the significance of differences between the two proportions. A *p*-value of less than 0.05 at a 95% confidence interval (95% CI) was considered statistically significant.

## 5. Conclusions

The overall laboratory confirmation of neonatal sepsis was 72%, predominantly caused by Gram-positive bacteria. Coagulase Negative *Staphylococci*, *Streptococcus agalactiae*, and *Staphylococcus aureus* were the three frequently isolated pathogens. The pathogen causing neonatal sepsis in our setting exhibited a high resistance rate to commonly used antibiotics, and about two-thirds of the pathogens were MDR strains; the presence of a high rate of MDR pathogens calls for a need to strengthen the laboratory diagnostic service and antibiotic stewardship practice.

## Figures and Tables

**Table 1 antibiotics-12-00767-t001:** Participants’ demographic and clinical characteristics (N = 658).

Variable		Frequency (n)	Percent (%)
Age	Median (IQR)	2 (1–8)	
	0–3	401	60.9
	≥4	257	39.1
Gender	Male	310	47.1
	Female	348	52.9
Place of birth	MRRH	325	49.4
	Other health facilities	333	50.6
Mode of delivery	Cesarian section	115	17.5
	Vaginal delivery	543	82.5
Gestation age at delivery *	<37 weeks	104	16.1
	≥37 weeks	541	83.9
Birth weight (grams)	Mean (SD)	2989.9 (±596.1)	
	<2500	97	14.7
	≥2500	561	85.3
Antibiotic use before #	Yes	104	16.4
	No	531	83.6
Onset of sepsis features	Early (≤3 days)	405	61.6
	Late (>3 days)	239	38.4
Clinical features	Median (IQR)	2 (2–3)	
	2	411	62.5
	>2	247	37.5
Fever (≥38 °C)	Yes	518	78.7
	No	140	21.3
Birth Asphyxia	Yes	50	7.6
	No	608	92.4
Empirical treatment	1st line regimen only	542	82.4
	1st and 2nd line regimens	116	17.6
Clinical outcome	Discharged	592	90.0
	Death	28	4.3
	Referred	38	5.8

* Total count was 645, and # was 635, IQR = interquartile range, MRRH = Mwananyamala Regional Referral Hospital, SD = Standard deviation, 1st line regimen = Ampicillin + Cloxacillin + Gentamycin, 2nd line regimen = Ceftriaxone.

**Table 2 antibiotics-12-00767-t002:** Laboratory confirmation and distribution of pathogens causing neonatal sepsis.

Variable		Frequency (n)	Percent (%)
Culture results	Bacterial growth	460	72.0
	No bacterial growth	179	28.0
Isolate type	Gram-positive	374	81.3
	Gram-negative	86	18.7
Isolated pathogen	CoNS	163	35.4
	*Streptococcus agalactiae*	99	21.5
	*Staphylococcus aureus*	94	20.4
	*Escherichia coli*	42	9.1
	*Klebsiella pneumoniae*	25	5.4
	*Enterococcus faecalis*	18	3.9
	*Pseudomonas aeruginosa*	9	2.0
	* Other GNB	10	2.2

CoNS = Coagulase-negative *Staphylococci*, GNB = Gram-negative bacteria, * Other GNB—*Enterobacter cloacae* (n = 3) and *Serratia marcescens* (n = 7).

**Table 3 antibiotics-12-00767-t003:** Proportion of antibiotic resistance in Gram-positive pathogens causing neonatal sepsis.

Antibiotic	CoNS (N = 161)	*S. aureus* (N = 92)	*S. agalactiae* (N = 98)	*E. faecalis* (N = 18)	Overall
	n (%)	n (%)	n (%)	n (%)	%
AMP	NA	NA	90 (91.8)	4 (22.2)	81.0
PEN	151 (94.4)	86 (93.5)	93 (94.9)	12 (66.7)	92.9
OXA	102 (63.4)	52 (56.5)	NA	NA	60.9
GEN	114 (70.8)	56 (60.9)	NA	10 (55.6)	66.4
CRO	NA	NA	77 (78.6)	NA	78.6
CIP	63 (39.1)	52 (56.5)	50 (51.0)	NA	47.0
ERY	138 (85.7)	79 (85.9)	85 (86.7)	NA	86.0
CLI	62 (38.5)	49 (53.3)	29 (29.6)	NA	39.9
CHL	39 (24.2)	32 (34.8)	10 (10.2)	NA	23.1
LNZ	92 (57.5)	66 (72.5)	NA	NA	62.9
SXT	114 (70.8)	74 (83.1)	NA	NA	74.3

N = The total number of isolates, n = the number of isolates resistant to an antibiotic, AMP = ampicillin, CHL = Chloramphenicol, CRO = ceftriaxone, CIP = ciprofloxacin, CLI = clindamycin, CoNS = Coagulase-negative *Staphylococci*, ERY = erythromycin, GEN = gentamicin, LNZ = linezolid, NA = not applicable, PEN = Penicillin, SXT = trimethoprim-sulfamethoxazole.

**Table 4 antibiotics-12-00767-t004:** Proportion of antibiotic resistance in Gram-negative pathogens causing neonatal sepsis.

Antibiotic	*E. coli* (N = 42)	*Klebsiella pneumoniae* (N = 24)	*P. aeruginosa* (N = 14)	* Other GNB (N = 10)	Overall
	n (%)	n (%)	n (%)	n (%)	%
AMP	37 (86.0)	23 (95.8)	NA	9 (90.0)	90.7
SXT	39 (92.9)	20 (83.3)	9 (100.0)	10 (100.0)	86.6
CHL	9 (21.4)	13 (54.2)	4 (44.4)	6 (60.0)	35.5
AMC	37 (88.1)	20 (83.3)	8 (88.9)	10 (100.0)	88.2
CIP	20 (47.6)	8 (33.3)	5 (55.6)	5 (50.0)	42.2
PIP	30 (71.4)	10 (41.7)	6 (66.7)	8 (80.0)	63.5
PIP/TZ	7 (16.7)	8 (33.3)	3 (33.3)	4 (40.0)	25.9
CAZ	30 (71.4)	17 (70.4)	8 (88.9)	10 (100.0)	76.5
CRO	21 (50.0)	12 (50.0)	7 (77.5)	8 (80.0)	53.3
GEN	19 (45.2)	12 (50.0)	5 (55.6)	6 (60.0)	46.7
AMK	9 (21.4)	6 (25.0)	2 (22.2)	4 (40.0)	24.7
MEM	9 (21.4)	10 (41.7)	1 (11.1)	3 (30.0)	27.1

N = The total number of isolates, n = the number of isolates resistant to an antibiotic, AMP = ampicillin, AMC = amoxicillin-clavulanic acid, AMK = amikacin, GNB = Gram-negative bacteria, CAZ = ceftazidime, CHL = Chloramphenicol, CIP = ciprofloxacin, CRO = ceftriaxone, GEN = gentamicin, MEM = meropenem, NA = not applicable, PIP = Piperacillin, PIPTAZ = Piperacillin/Tazobactam, SXT = trimethoprim-sulfamethoxazole. * Other GNB includes *Enterobacter cloacae* (n = 3) and *Serratia marcescens* (n = 7).

**Table 5 antibiotics-12-00767-t005:** Empirical treatment and antibiotic susceptibility pattern.

Empirical Therapy	Result of Antibiotic Susceptibility		
	Total Susceptibility Tests	Appropriate * n (%)	Inappropriate n (%)
Overall 1st line regimen	454	198 (43.6)	256 (56.4)
1st line only regimen	372	165 (44.4)	207 (55.6)
2nd line regimen	35	9 (25.7)	26 (74.3)

* Appropriate = when the pathogen was susceptible to one or more of the antibiotics in empirical therapy, 1st line regimen = Ampicillin + Cloxacillin + Gentamycin, 2nd line regimen = Ceftriaxone.

**Table 6 antibiotics-12-00767-t006:** Proportion of MDR strains of bacteria pathogens causing neonatal sepsis.

Variable		Frequency #	Antibiotic Classes Resisted				MDR Strains n (%)	*p*-Value
			1–2	3	4	5		
Isolate type	Gram-positive	351	106	92	86	65	243(69.2)	0.334
	Gram-negative	85	21	13	13	38	64(75.3)	
Isolated pathogen	*Streptococcus agalactiae*	98	58	34	6	0	40 (40.8)	
	CoNS	161	33	47	51	30	128 (79.5)	
	*Staphylococcus aureus*	92	15	11	29	35	75 (81.5)	
	*Klebsiella pneumoniae*	24	10	3	3	8	14 (58.3)	
	*Escherichia coli*	42	10	6	10	16	32 (76.2)	
	*Pseudomonas aeruginosa*	9	1	3	0	5	8 (88.9)	
	* Other GNB	10	0	1	0	9	10 (100.0)	
Total		436	127	105	99	103	307 (70.4)	

# MDR strains were determined for 436 isolates excluding those tested with less than four classes of antibiotics, MDR = Multidrug-resistant, * Other GNB = Gram-negative bacteria (*Enterobacter cloacae* (n = 3) and *Serratia marcescens* (n = 7)).

## Data Availability

The datasets used and analysed during the current study are available from the corresponding author upon reasonable request.

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
