# Peer review of "Bacterial Aetiology of Neonatal Sepsis and Antimicrobial Resistance Pattern at the Regional Referral Hospital, Dar es Salam, Tanzania; A Call to Strengthening Antibiotic Stewardship Program"

_antibiotics, 2023, doi:10.3390/antibiotics12040767_

Round 1
Reviewer 1 Report
The paper in of interest. Since in case of neonatal sepsis blood culture results are not immediately available (and in general there is a non-negligible percentage of false negatives), the authors should indicate in the discussion what is the empirical therapy adopted in their hospital and whether it was changed based on the results of their study.
Author Response
Reviewer 1
The paper in of interest. Since in case of neonatal sepsis blood culture results are not immediately available (and in general there is a non-negligible percentage of false negatives), the authors should indicate in the discussion what is the empirical therapy adopted in their hospital and whether it was changed based on the results of their study.
Response: We have included information about empirical therapy in the background (lines 59-63), in results (lines 82-84, 117-122 and tables 1 and 5), and in discussion (lines219-225)
Reviewer 2 Report
The manuscript “bacterial etiology of neonatal sepsis and antimicrobial resistance pattern at the regional referral hospital Dar es Salaam Tanzania. A call to strengthening antibiotic stewardship program” reports important and quiet alarming rates of resistance, with a potential important clinical implication.
However, before publication is warranted, several issues have to be addressed.
General:
Authors choose to only report antibiotic resistance patterns and did not report specific data of empirical antibiotic therapy choices and clinical outcome of the patients. By doing so, an important opportunity has been missed to show in how many cases empiric antibiotic therapy did match the resistance pattern of the cultured pathogen and if there is an association between antibiotic resistance and clinical outcome. Adding these data will substantially strengthen the manuscript.
Also data of intravenous central venous lines/canula’s has not been reported, which can be helpful to interpret the data, especially since a high proportion of CoNS has been reported.
Methods:
4.4.1
“API 20 E and API 20-NE were also used” can authors be more specific in which cases?
4.4.2. antibiotic susceptibility testing.
For gram positive bacteria authors choose not to test several antibiotic inclusing cephalosporins, amoxi-clav and vancomycine. Can you please explain why the choice has been made not to test these antibiotics? These choices limit the interpretation of the data.
Results;
Minor remark: 2.1 line 4 “on the other hand” ; seems misplaced in this sentence
Table 1: outcome (survival or mortality) is not included in the table
Table 3. By reporting the percentage resistance for each antibiotic separately, information is missing. A supplemental data file with the susceptibility/resistance patterns for each individual bacteria can be helpful. In case these data are available and if data is available concerning empiric antibiotic therapy, one can calculate the proportion of patients with appropriate and inappropriate empiric therapy. Important information of e.g. vancomycine resistance is missing because of methodological reasons, see above.
2.4
Concerning MDR stains the proportions are mentioned per pathogen, but not the distribution of the antibiotic classes for which resistance occurred in these pathogens. This can also be tackled by adding a supplemental file with the susceptibility/resistance patterns for each bacteria.
Discussion:
Please address the issue of contamination while discussion the high proportion of CoNS bacteremia.
Our study found….etc: it is not very useful to discuss the percentage of resistance of all isolates, because the discussion of resistance patterns is only relevant for specific bacteria, because differences in intrinsic and acquired resistance patterns and different mechanisms of resistance are pathogen specific. The discussion of gram positive and gram negative resistance patterns has also limitations for the same reason.
“irrational use of antibiotics in empiric treatment, which is the practice in our setting” : data on empiric antibiotic therapy were not presented. Are these data available? These data are also very important to estimate which stewardship interventions/programs could be effective in this setting.
Author Response
Reviewer 2
The manuscript “bacterial etiology of neonatal sepsis and antimicrobial resistance pattern at the regional referral hospital Dar es Salaam Tanzania. A call to strengthening antibiotic stewardship program” reports important and quiet alarming rates of resistance, with a potential important clinical implication.
However, before publication is warranted, several issues have to be addressed.
General: Authors choose to only report antibiotic resistance patterns and did not report specific data of empirical antibiotic therapy choices and clinical outcome of the patients. By doing so, an important opportunity has been missed to show in how many cases empiric antibiotic therapy did match the resistance pattern of the cultured pathogen and if there is an association between antibiotic resistance and clinical outcome. Adding these data will substantially strengthen the manuscript. Also, data of intravenous central venous lines/canula’s has not been reported, which can be helpful to interpret the data, especially since a high proportion of CoNS has been reported.
Response: Thanks for the valid comment; we have included the empirical antibiotics used and the clinical outcome of the patients in Table 1. In addition, we have added Table 5 to match the empirical treatment and resistance pattern of the culture and susceptibility-tested pathogens. We have reported the association between antibiotic resistance and clinical outcome (line 133). Unfortunately, we did not record the neonates who had intravenous central lines prior to neonatal sepsis
Methods
4.4.1: “API 20 E and API 20-NE were also used” can authors be more specific in which cases?
Response: Thanks for the comment; We have included the information indicating that We used API 20E and API 20-NE to confirm the identity when the conventional biochemical test provided an inconclusive result (lines 282-283)
4.4.2. antibiotic susceptibility testing.
For gram positive bacteria authors choose not to test several antibiotic inclusing cephalosporins, amoxi-clav and vancomycine. Can you please explain why the choice has been made not to test these antibiotics? These choices limit the interpretation of the data.
Response: We have added ceftriaxone to the list of antibiotics tested for Gram-positive, specifically for Streptococci (line 300 and table 3). As we used CLSI guideline for AST, most of the cephalosporins we used and amoxiclav are not recommended for the Gram-positive bacteria we isolated. Vancomycin was missing because of methodologic reasons; it requires the determination of minimum inhibition concentration.
Results;
Minor remark: 2.1 line 4 “on the other hand”; seems misplaced in this sentence
Response: The sentence has been rephrased by deleting “on the other hand”(line 76)
Table 1: outcome (survival or mortality) is not included in the table
Response: We have added to Table One the patient outcome
Table 3. By reporting the percentage resistance for each antibiotic separately, information is missing. A supplemental data file with the susceptibility/resistance patterns for each individual bacteria can be helpful. In case these data are available and if data is available concerning empiric antibiotic therapy, one can calculate the proportion of patients with appropriate and inappropriate empiric therapy. Important information of e.g. vancomycine resistance is missing because of methodological reasons, see above.
Response: We appreciate the comment. we have added Table 5 which shows the proportion of patients with appropriate and inappropriate empirical treatment among those culture-positive and AST performed.
2.4: Concerning MDR stains the proportions are mentioned per pathogen, but not the distribution of the antibiotic classes for which resistance occurred in these pathogens. This can also be tackled by adding a supplemental file with the susceptibility/resistance patterns for each bacteria.
Response: Thanks for the comment: We have added to Table 6 the number of antibiotic classes in which resistance occurred for each pathogen.
Discussion:
Please address the issue of contamination while discussion the high proportion of CoNS bacteremia.
Response: We have added more information in the discussion concerning CoNS bacteremia and contamination (lines 174-192)
Our study found….etc: it is not very useful to discuss the percentage of resistance of all isolates, because the discussion of resistance patterns is only relevant for specific bacteria,because differences in intrinsic and acquired resistance patterns and different mechanisms of resistance are pathogen specific. The discussion of gram positive and gram negative resistance patterns has also limitations for the same reason.
Response: Thanks for the comment; we have revised the paragraph by adding more information discussing the resistance patterns for some specific bacteria (lines 206-215)
“irrational use of antibiotics in empiric treatment, which is the practice in our setting”: data on empiric antibiotic therapy were not presented. Are these data available? These data are also very important to estimate which stewardship interventions/programs could be effective in this setting.
Response: We have added data and information on the empiric therapy in the background (lines 59-63), in results (lines 82-84, 117-122 and tables 1 and 5), and in discussion (lines 219-225)
Reviewer 3 Report
Review Antibiotics Journal
Manuscript
Bacterial Aetiology of Neonatal Sepsis and Antimicrobial Re- 2 sistance Pattern at the Regional Referral Hospital, Dar es Salam, Tanzania. A Call to Strengthening Antibiotic Stewardship Program
Antibiotic stewardship is important issue in medical practice. Collection and management of antibiotic resistance and MDR are interesting for the local antibiotic stewardship programmes. There are les known the antibiotic resistance patterns in neonates.
The references of the article are too old, it must updated. Mainly, The WHO guideline recommendations for antibiotic treatment are changed since 2015, after your reference.
The introduction has outdated information. It should be consistently revised.
A clear formulation of the hypothesis and objective of the study are required.
The methodology is too thin. Firstly, there are not presented the criteria for neonate sepsis that have supported the diagnostic.
Additionally, other information should be significant for the results:
-the localization of infection related neonate sepsis
-the severity of sepsis – pointing the organs associated failures
-witch other biological samples were positive with the same germs as haemocultures
-maternal risk factors- infections or colonization with germs with antibiotic resistance, as antibiotic use in the proximal time before birth.
The most frequent aetiology is CoNS, but it not convincing that it was a real sepsis or transitory bacteraemia or contamination in a febrile neonate.
The description of statistic methods is too summary. The statistic analyse is not consistent. It should be supported by the correlations that could predict the antibiotic resistance risk.
Overall, I consider that the manuscript is not appropriate for the standards of Antibiotics Journal, but could be revised and resubmitted to an other journal.
Author Response
Reviewer 3
Manuscript: Bacterial Aetiology of Neonatal Sepsis and Antimicrobial Re- 2 sistance Pattern at the Regional Referral Hospital, Dar es Salam, Tanzania. A Call to Strengthening Antibiotic Stewardship Program
Antibiotic stewardship is important issue in medical practice. Collection and management of antibiotic resistance and MDR are interesting for the local antibiotic stewardship programmes. There are les known the antibiotic resistance patterns in neonates. The references of the article are too old, it must updated. Mainly, The WHO guideline recommendations for antibiotic treatment are changed since 2015, after your reference. The introduction has outdated information. It should be consistently revised.
Response: Thanks for the comment; we have updated the references, especially the WHO guideline (Ref. 11).
A clear formulation of the hypothesis and objective of the study are required.
Response: Thanks for the comment; the objective of our study is stated in the last paragraph of the introduction (lines 65-70)
The methodology is too thin. Firstly, there are not presented the criteria for neonate sepsis that have supported the diagnostic. Additionally, other information should be significant for the results: -the localization of infection related neonate sepsis
Response: The criteria for neonatal sepsis are indicated in the methodology section (lines 251-258), which warranted for blood culture, although there were possibilities of having localised infection
-the severity of sepsis – pointing the organs associated failures
Response: Unfortunately, we did not categorise the severity of the infection
-witch other biological samples were positive with the same germs as haemocultures
Response: Unfortunately, we processed only the blood for culture.
-maternal risk factors- infections or colonization with germs with antibiotic resistance, as antibiotic use in the proximal time before birth.
Response: The study focused on neonatal infection; we did not screen for maternal infection or colonisation.
The most frequent aetiology is CoNS, but it not convincing that it was a real sepsis or transitory bacteraemia or contamination in a febrile neonate.
Response: We agree that CoNS may not be the causative agent instead transitory bacteraemia or contamination. We have added more information about CoNS in the discussion ((lines 174-192)
The description of statistic methods is too summary. The statistic analyse is not consistent. It should be supported by the correlations that could predict the antibiotic resistance risk.
Response: Thanks for the comment. Our study was a descriptive cross-sectional design; we did not correlate the predictors of antibiotic resistance; the aim was to determine the aetiology and antibiotic resistance pattern
Overall, I consider that the manuscript is not appropriate for the standards of Antibiotics Journal, but could be revised and resubmitted to an other journal.
Response: Thanks for the comment; we think that our study has important information on antibiotic resistance which is within the scope of the journal
Round 2
Reviewer 2 Report
The authors adequately addressed the reviewers comments
Author Response
Comment
The authors adequately addressed the reviewers' comments
Response: Thanks for the positive comment. In addition, we have reviewed the manuscript for spell check and made a few corrections.
Reviewer 3 Report
It should comment the results of the study reporting to the up-to date recommendation of the current guidelines, as:
https://www.sccm.org/SurvivingSepsisCampaign/Guidelines/Pediatric-Patients
https://journals.lww.com/pccmjournal/Fulltext/2020/02000/Surviving_Sepsis_Campaign_International_Guidelines.20.aspx
Author Response
Comments
It should comment the results of the study reporting to the up-to date recommendation of the current guidelines, as:
https://www.sccm.org/SurvivingSepsisCampaign/Guidelines/Pediatric-Patients
https://journals.lww.com/pccmjournal/Fulltext/2020/02000/Surviving_Sepsis_Campaign_International_Guidelines.20.aspx
Response: Thanks for the comments and the link to the current guideline. We have used the reference (reference 11) to replace the old one. In addition, we have included the recommendation of the current guideline in the introduction (Lines 47-50)